# The global burden of chromoblastomycosis

**Daniel Wagner C. L. Santos** [1,2]\*, **Conceição de Maria Pedrozo e Silva de Azevedo** [3,4], **Vania Aparecida Vicente** [5], **Flávio Queiroz-Telles** [6], **Anderson Messias Rodrigues** [7], **G. Sybren de Hoog** [5,8], **David W. Denning** [9,10], **Arnaldo Lopes Colombo** [1,2]\*

**1** Special Mycology Laboratory—LEMI, Division of Infectious Diseases, Federal University of São Paulo, São Paulo, SP, Brazil, **2** Division of Infectious Diseases, Federal University of São Paulo, São Paulo, SP, Brazil, **3** Department of Medicine, Federal University of Maranhão, São Luís, MA, Brazil, **4** Post-graduation Program of Health Science, Federal University of Maranhão, São Luís, MA, Brazil, **5** Microbiology, Parasitology, and Pathology Post Graduation Program, Department of Pathology, Federal University of Paraná, Curitiba, Brazil, **6** Department of Public Health, Hospital de Clínicas, Federal University of Paraná, Curitiba, Paraná, Brazil, **7** Laboratory of Emerging Fungal Pathogens, Department of Microbiology, Immunology, and Parasitology, Federal University of São Paulo, São Paulo, SP, Brazil, **8** Center of Expertise in Mycology, Radboud University Medical Center/CWZ, Nijmegen, The Netherlands, **9** Global Action Fund for Fungal Infections, Geneva, Switzerland, **10** Manchester Fungal Infection Group, Core Technology Facility, The University of Manchester, Manchester, United Kingdom

\* danielinfectologista@gmail.com (DWCLS); arnaldolcolombo@gmail.com (ALC)

**Data Availability Statement:** All relevant data are within the manuscript and its Supporting Information files.

## Abstract

### Background

Chromoblastomycosis (CBM), represents one of the primary implantation mycoses caused by melanized fungi widely found in nature. It is characterized as a Neglected Tropical Disease (NTD) and mainly affects populations living in poverty with significant morbidity, including stigma and discrimination.

### Methods and findings

In order to estimate the global burden of CBM, we retrospectively reviewed the published literature from 1914 to 2020. Over the 106-year period, a total of 7,740 patients with CBM were identified on all continents except Antarctica. Most of the cases were reported from South America (2,619 cases), followed by Africa (1,875 cases), Central America and Mexico (1,628 cases), Asia (1,390 cases), Oceania (168 cases), Europe (35 cases), and USA and Canada (25 cases). We described 4,022 (81.7%) male and 896 (18.3%) female patients, with the median age of 52.5 years. The average time between the onset of the first lesion and CBM diagnosis was 9.2 years (range between 1 month to 50 years). The main sites involved were the lower limbs (56.7%), followed by the upper limbs (19.9%), head and neck (2.9%), and trunk (2.4%). Itching and pain were reported by 21.5% and 11%, respectively. Malignant transformation was described in 22 cases. A total of 3,817 fungal isolates were cultured, being 3,089 (80.9%) *Fonsecaea* spp., 552 (14.5%) *Cladophialophora* spp., and 56 *Phialophora* spp. (1.5%).

### Conclusions and significance

This review represents our current knowledge on the burden of CBM world-wide. The global incidence remains unclear and local epidemiological studies are required to improve these

**Funding:** The authors received no specific funding for this work.

**Competing interests:** The authors have declared that no competing interests exist.

data, especially in Africa, Asia, and Latin America. The recognition of CBM as NTD emphasizes the need for public health efforts to promote support for all local governments interested in developing specific policies and actions for preventing, diagnosing and assisting patients.

## Author summary

Chromoblastomycosis (CBM), represents one of the primary implantation mycoses caused by melanized fungi widely found in nature. It is characterized as a Neglected Tropical Disease and mainly affect populations living in poverty with significant morbidity, including stigma and discrimination. The global incidence of CBM remains unclear because this mycosis is not a mandatory notifiable disease and most of the literature consists of case reports or small series incompletely characterized. Although several authors suggest that the CBM global burden may be comparable to mycetoma, its geographic distribution and incidence rates in different endemic areas have never been widely characterized in the medical literature. We retrospectively conducted a comprehensive systematic review of all medical literature published between 1914 and 2020 to better characterize the prevalence rates, geographic distribution, and clinical aspects of CBM in all continents. All reviewed data were not a substitute for high quality epidemiological study or comprehensive surveillance but do provide an approximation of the burden by country. Information generated corroborate the WHO recognition of CBM as a NTD and provides helpful support for all local governments interested in developing specific policies and actions for preventing, diagnosing and assisting patients with CBM.

## Introduction

Chromoblastomycosis (CBM), together with mycetoma, represents one of the primary implantation mycoses caused by melanized or black fungi widely found in nature that may infect agricultural workers after transcutaneous inoculation during their daily activities [1–4]. Chromoblastomycosis is primarily an occupational disease associated with a considerable social stigma and severe personal and family socioeconomic consequences [1,3,5,6]. It is mainly caused by *Fonsecaea* spp., followed by *Cladophialophora*, *Phialophora*, and *Rhinocladiella*. The genera *Fonsecaea* includes three closely related siblings represented by *F. pedrosoi*, *F. monophora*, and *F. nubica*. The genus *Cladophilophora* spp. contains two related siblings: *C. carrionii* that may be found in clinical samples and nature, whereas *C. yegresii* is exclusively found in the environment [1,7–12]. These agents present some peculiarities in terms of geographic distribution and ecological niches. The clinical manifestations and therapeutic response of the patients differs by infecting fungus.

  CBM is nowadays characterized as a Neglected Tropical Disease (NTD) because (a) it mainly affects populations living in poverty causing significant morbidity and mortality–including stigma and discrimination; (b) it is mostly found in tropical and sub-tropical areas; (c) it may be controlled or eradicated by applying one or more of the five public health strategies adopted by the Department for Control of NTDs; (d) it has been neglected by research when it comes to developing new diagnostics, medicines, and other control tools [1,3–5]. The process of recognizing CBM as NTD began at the meeting held in São Luís, state of Maranhão, Brazil, in 2011, when the disease's centenary was celebrated. After an application by the Global

Action Fund for Fungal Infections with support from the governments of Brazil and Madagascar, World Health Organization (WHO) incorporated CBM into the NTD portfolio in category B in 2017, together with mycetoma and other deep mycoses [1,7].

In most endemic areas, health services do not have professionals trained in the early diagnosis and clinical management of CBM. Skills in skin biopsy, direct microscopy, histopathology with fingal stains fungal culture is often lacking. Effective antifungal treatment rarely included in universal health coverage and government insurance. Long term itraconazole at 400mg daily or variable terbinafine dose are not be available in many countries and is expensive and requires monitoring [1,3,6–9]. Patients are usually diagnosed after several years of clinical manifestations, and medication is unavailable or unaffordable, two factors that may increase the risk of sequelae and further social stigma [1,3]. In addition, CBM in some patients is complicated by continuous bacterial co-infection and later neoplastic transformation of the CBM lesions into epidermoid carcinoma may occur [1,3]

The global incidence of CBM remains unclear. Only a few population epidemiology studies have been done. This mycosis is not a mandatory notifiable disease and most of the literature consists of case reports or small series incompletely characterized. Although several authors suggest that the CBM global burden may be comparable to mycetoma, its geographic distribution and incidence rates in different endemic areas have never been widely characterized in the medical literature.

We conducted a comprehensive systematic review of all medical literature published between 1914 and 2020 to characterize better the prevalence rates and geographic distribution of CBM in all continents. Data generated in this paper corroborate the WHO recognition of CBM as a NTD and provides helpful support for all local governments interested in developing specific policies and actions for preventing, diagnosing, and assisting patients with CBM [1,5–7,13,14].

## Methods

Our plan for literature review included the selection of all articles addressing the epidemiology of chromoblastomycosis in the world that were published in four different languages (English, Spanish, French and Portuguese) between 1914 and 2020 and listed in the PubMed (https://www.ncbi.nlm.nih.gov/pubmed/) and Bireme (http://portal.revistas.bvs.br/) with access to "LILACS", "IBECS", "MEDLINE", "Cochrane Library" and "SciELO" databases. The terms used to select papers included "chromoblastomycosis", "chromomycosis", "neglected mycoses", "subcutaneous mycoses" or "implantation fungal infections". Letters to the editor and abstracts available published in congress or conferences were also searched and identified. The literature review was complemented by reviewing the reference lists of all studies selected to be sure that we did not miss any relevant references. Due to the large number of single case reports published in some highly endemic countries, papers from Mexico, Brazil, Venezuela, Colombia, Madagascar, India, China, Japan, and Australia were only included if they reported at least 5 patients. Review papers were selected only to find references to original papers to avoid case duplication [2,15]. All papers in this comprehensive review were able to meet the main diagnostic criteria of CBM: presence of dark pigmented and thick-walled muriform cells in a biological sample. Whenever there was doubt about this finding, the paper was excluded from the analysis.

Epidemiological and clinical data of all cases of CBM were collected using a standard clinical form. The variables that were systematically assessed along the literature review included year and country of publication, the number of cases, the period of cases collection, age, gender, history of cutaneous trauma and previous agricultural work, time from onset of symptoms to diagnosis (years), symptoms, clinical pattern of the lesions and severity of the disease,

malignant transformation and clinical management (physical methods such as surgery, thermotherapy, laser therapy, and photodynamic therapy; antifungal drugs with itraconazole, terbinafine, iodide, flucytosine) [1,3].

To determine the prevalence rate of CBM in each country, we used the method described by Van de Sande [2]. The number of reported cases along each year in all countries was divided by the total population of each country in the selected period. Population data for each country in each collection period was extracted from the website www.indexmundi.com/facts/indicators/SP.POP.TOTL/compare#country=ma) [2]. As an example 71 cases of CBM were reported between 1978 and 1993 in Sri Lanka, with a mean of 4.73 cases/year. The average population of this country in this period was 16,283,921 inhabitants. In this case, the prevalence of CBM in Sri Lanka was defined as 0.29 cases per 1 million inhabitants.

## Results

Our review identified a total of 208 articles that were published in English (119 articles), Spanish (42 articles), French (39 articles), and Portuguese (8 articles), accounting for 7,740 cases of CBM on all continents except Antarctica. The main characteristics of CBM are illustrated by countries and continents, as summarized in Tables 1 and S1. The worldwide distribution and prevalence of CBM cases are shown in Fig 1.

**Table 1. Epidemiology and main clinical aspects of 7,740 cases of chromoblastomycosis documented in 5 different continents.**

| Variables | South America (n = 2,619) | Central America, The Caribbean, and Mexico (n = 1,628) | Africa (n = 1,875) | Asia (n = 1,390) | Europe (n = 35) | USA and Canada (n = 25) | Oceania (n = 168) | Total (n = 7,740) |
|---|---|---|---|---|---|---|---|---|
| Age (years), (mean, range) | 57.1 (12y-93y) | 53.3 (9y-90y) | 47.9 (2y-73y) | 49.7 (7y-90y) | 60.9 (17y-85y) | 55.7 (19y-79y) | 53.8 (19y-91y) | 52.5 (2y-93y) |
| Male/Female n (%) | 1,237 (87.4%) / 178 (12.6%) | 802 (76.7%) /243 (23.3%) | 1,338 (83.6%) / 263 (16.4%) | 463 (71.6%) / 183 (28.4%) | 27 (77.2%) / 8 (22.8%) | 22 (88%) / 3 (12%) | 133 (88%) /18 (12%) | 4,022 (81.7%) / 896 (18.3%) |
| Rural occupation | 927 | 642 | 129 | 162 | 2 | 13 | 20 | 1,895 |
| History of trauma | 276 | 117 | 20 | 116 | 18 | 3 | 18 | 568 |
| Delay between onset and diagnosis (mean, years) | 10.8 (1mo - 50y) | 13.4 (2mo-28y) | 8.7 (4mo-31y) | 8.3 (1mo-40y) | 13.6 (3mo-31y) | 4.8 (2m-20y) | 8.02 (1mo-30y) | 9.2 (1mo-50y) |
| **Sites of lesions** | | | | | | | | |
| Lower limbs | 1,021 | 472 | 1,340 | 308 | 16 | 6 | 34 | 3,197 |
| Upper limbs | 301 | 288 | 196 | 214 | 11 | 16 | 94 | 1,120 |
| Face, head, neck | 47 | 13 | 20 | 73 | 1 | 2 | 2 | 158 |
| Trunk | 54 | 42 | 32 | 50 | 2 | 0 | 0 | 180 |
| Unusual sites | 11 | 2 | 4 | 39 | 4 | 1 | 0 | 61 |
| **Types of lesions** | | | | | | | | |
| Plaque | 248 | 39 | 153 | 109 | 4 | 11 | 1 | 565 |
| Verrucous | 329 | 162 | 131 | 76 | 0 | 7 | 5 | 710 |
| Tumoral | 87 | 6 | 558 | 9 | 1 | 0 | 2 | 663 |
| Nodular | 84 | 46 | 117 | 24 | 1 | 5 | 8 | 285 |
| Scarring | 37 | 1 | 32 | 2 | 1 | 0 | 2 | 75 |
| Ulcer | 47 | 7 | 27 | 16 | 1 | 7 | 0 | 105 |
| **Etiologic agents** | 864 | 703 | 1,406 | 750 | 30 | 21 | 43 | 3,817 |
| *Fonsecaea* spp. | 759 (87.9%) | 676 (96.2%) | 1,017 (72.3%) | 569 (75.8%) | 24 (80%) | 15 (71.5%) | 29 (67.4%) | 3,089 (80.9%) |
| *Cladophialophora* spp. | 82 (9.5%) | 11 (1.6%) | 376 (26.8%) | 68 (9%) | 1 (3.3%) | 0 | 14 (32.6%) | 552 (14.5%) |
| *Phialophora* spp. | 12 (1.4%) | 5 (0.7%) | 8 (0.6%) | 24 (3.2%) | 1 (3.3%) | 6 (28.5%) | 0 | 56 (1.5%) |
| *Rhinocladiella* spp. | 5 (0.6%) | 1 (0.1%) | 0 | 2 (0.3%) | 1 (3.3%) | 0 | 0 | 9 (0.2%) |
| Other and not identified agent | 6 (0.6%) | 10 (1.4%) | 4 (0.3%) | 87 (11.7%) | 3 (10.1%) | 0 | 0 | 110 (2.9%) |

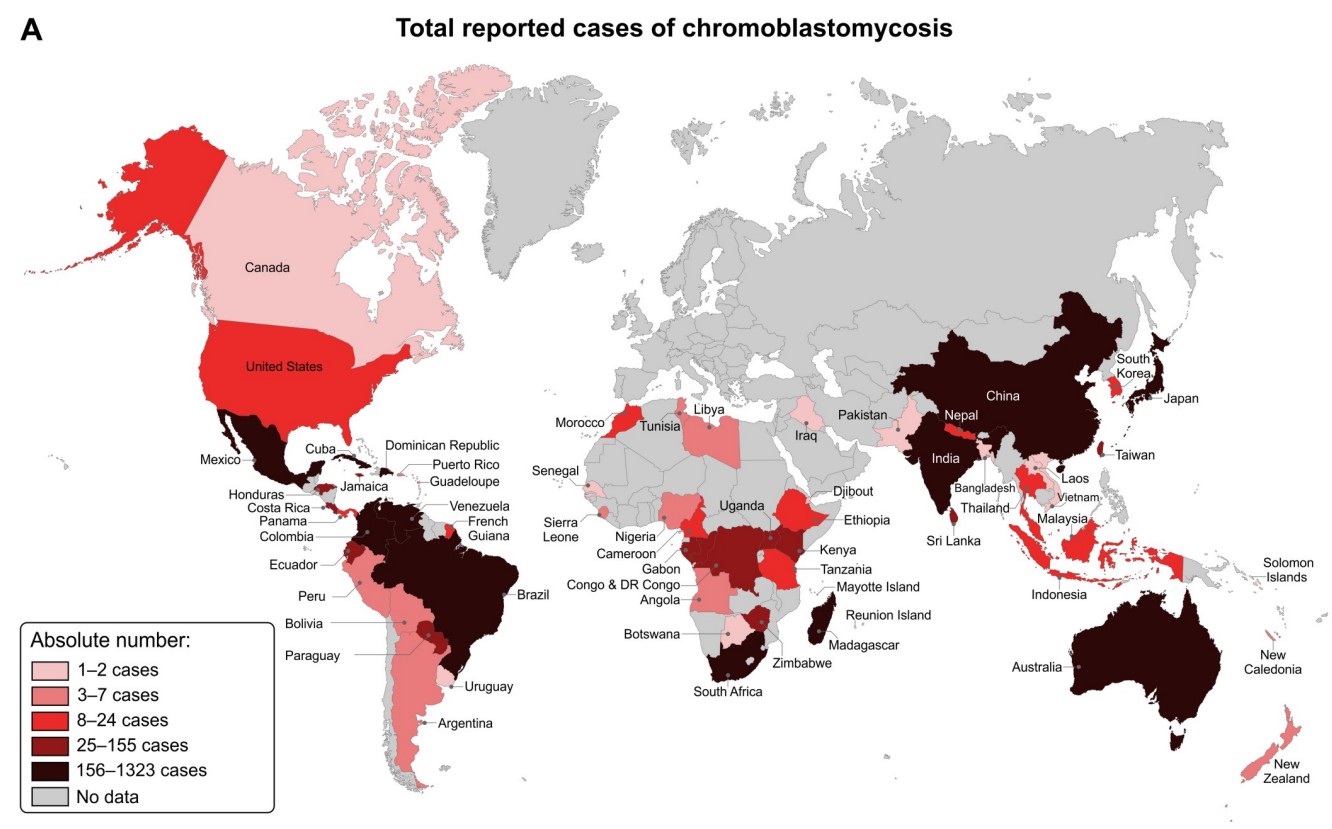

**A** **Total reported cases of chromoblastomycosis**

Absolute number:
- 1–2 cases
- 3–7 cases
- 8–24 cases
- 25–155 cases
- 156–1323 cases
- No data

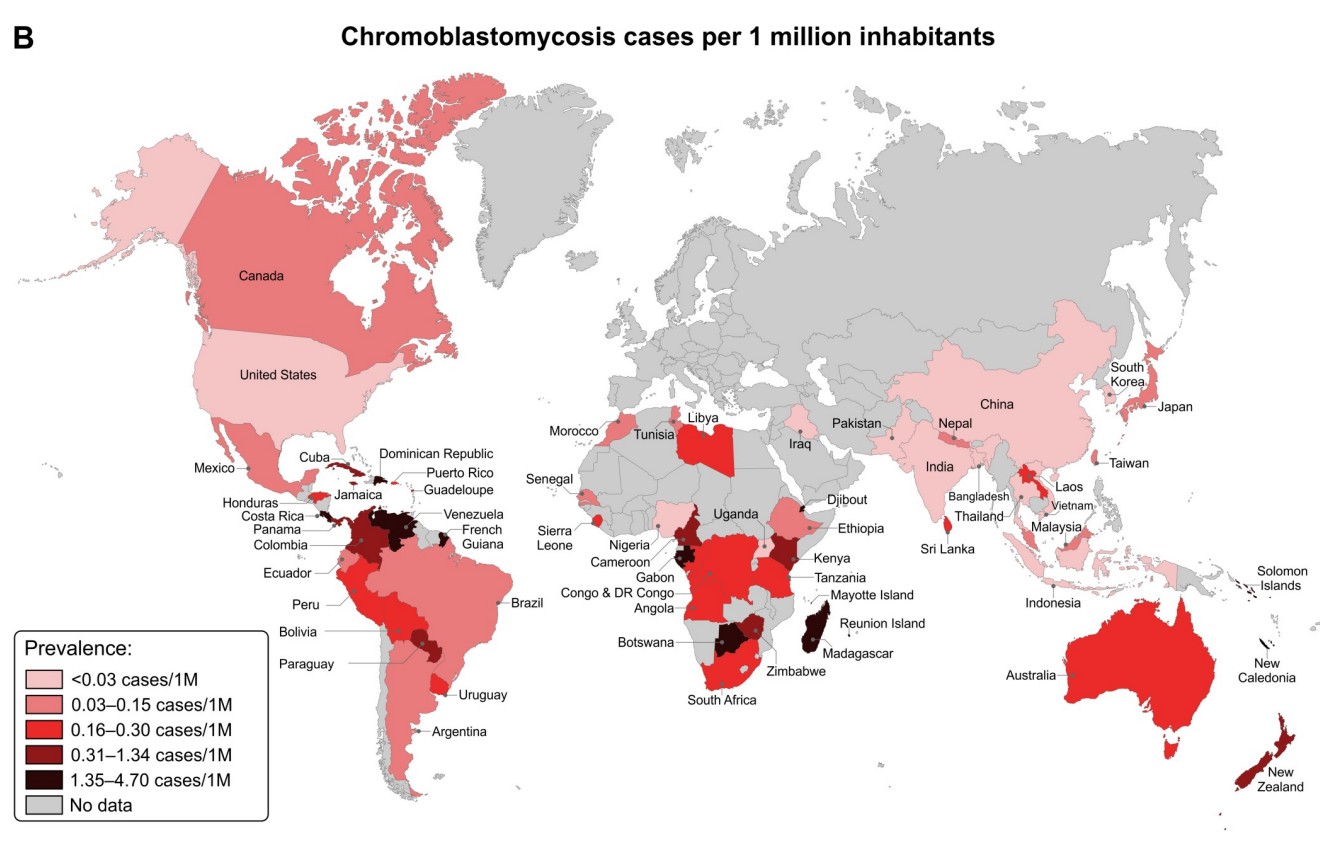

**B** **Chromoblastomycosis cases per 1 million inhabitants**

Prevalence:
- <0.03 cases/1M
- 0.03–0.15 cases/1M
- 0.16–0.30 cases/1M
- 0.31–1.34 cases/1M
- 1.35–4.70 cases/1M
- No data

**Fig 1. Prevalence and absolute number of reported cases of chromoblastomycosis.** The world map was created, edited, and colored using the vector graphics editor Corel Draw X8. Public domain link to map base layer used in creating this figure: https://commons.wikimedia.org/wiki/File: BlankMap-World.svg.

## Chromoblastomycosis in Central America, the Caribbean and Mexico

A total of 26 articles were identified which described 1,628 patients who were documented in Mexico (603 cases) [16], Dominican Republic (450 cases) [17], Cuba (319 cases) [18–25], Costa Rica (153 cases) [26–30], Honduras (52 cases) [31–34], Jamaica (31 cases) [35], Panama (8 cases) [36–39], Puerto Rico (7 cases) [40], and Guadalupe Islands (5 cases) [41]. Data from the Dominican Republic relies on a single publication from Isa-Isa, who mentioned 450 cases since 1966 without further details [17].

The authors described 802 male and 243 female patients, with Mexico the country with the lowest male: female ratio (1.95:1). They mention exposure to rural activities and history of trauma for 61.5% (642 out 1,043 cases) and 48% (117 out 244 cases), respectively. Histological findings were described for 269 (16.5%) patients. Cultures from the patient's lesions allowed the isolation of 703 clinical isolates, being *Fonsecaea* spp. (676; 96.2%) the most frequent isolated agent with geographic distribution throughout Central America [16,19,20,30,35,40–42]. *Cladophialophora* spp. was found only in Mexico, Cuba, and Costa Rica [16,20,21,30]. A molecular identification study conducted in Mexico found only *F. pedrosoi* in all nine samples of *Fonsecaea* spp. isolated in culture [42]. The highest prevalence rates of CBM per 1 million inhabitants were observed in Costa Rica, Dominican Republic, Panama, and Guadalupe Island [16–41]. The primary epidemiologic and clinical data of all CBM cases reported in the region were summarized in Tables 1 and S1.

## Chromoblastomycosis in South America

A total of 51 articles described 2,619 patients that were reported in Venezuela (1,167 cases) [43–49], Brazil (1,143 cases) [50–70], Colombia (167 cases) [71–76], Paraguay (82 cases) [77–81], Ecuador (34 cases) [82,83], French Guiana (11 cases) [84–86], Peru (7 cases) [87–89], Argentina (4 cases) [90,91], Bolivia (3 cases) [92], and Uruguay (1 case) [93]. In Brazil, most cases came from the Amazon region (states of Pará, Rondonia, and Amazon), Maranhão state (northeast region), in addition to the Central-West Region (Mato Grosso) and South Regions (states of Paraná and Rio Grande do Sul) [50–54,56–58,62,68–70]. Venezuela is an important endemic area in South America, and the disease is found throughout the country, with the states of Falcón, Lara, and Zulia standing out [44,45,48]. The Falcón region is responsible for 55% of the cases described in the country [48]. Data from Peru, Bolivia, Argentina, and Uruguay are scarce, as most cases were divulgated only in local meetings and conferences or published in non-indexed publications [87–93]. Biagini et al. (1982) reported that from 1929 to 1982, 11 cases of the disease were recorded in Argentina. According to Ricardo Negroni (personal communication), there were 2 to 3 cases per year in most medical centers in Buenos Aires [91].

The authors described 1,237 male and 178 female patients. Venezuela, as mentioned in several publications, had the lowest male:female rate [43,44,47]. They mentioned the previous exposition to rural activities and history of trauma for 74.5% (927 out 1,244 cases) and 53.1% (276 out 520 cases), respectively. Histological findings were described for 555 (21.2%) patients. Cultures from the patient's lesion allowed the isolation of 864 isolates. *Fonsecaea* spp. (759; 87.9%) was the most frequently found agent, followed by *Cladophialophora* spp. (82; 9.5%). The latter agent is typical of semi-arid rural areas of Venezuela and has been found sporadically in Brazil, Paraguay, Ecuador, and Peru [43–46,48,49,60,63,77,83,89]. The report by Richard-Yegres and Yegres accounted for 900 cases of CBM in Venezuela, with 490 (54.5%) in the state of Falcón, caused almost exclusively by *C. carrionii* [48]. Thus, the real number of cases

## A  Distribution patterns of *Fonsecaea* agents of chromoblastomycosis

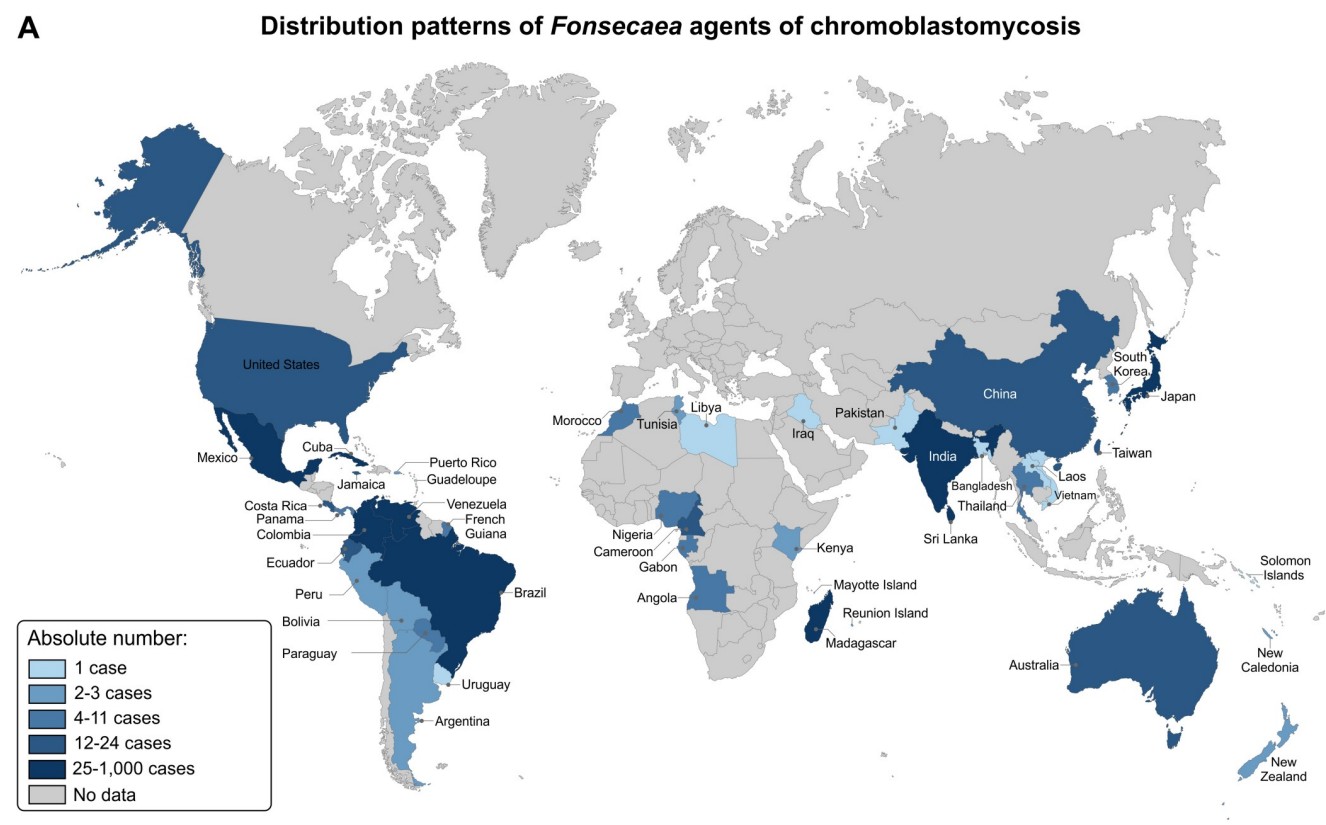

## B  Distribution patterns of *Cladophialophora* agents of chromoblastomycosis

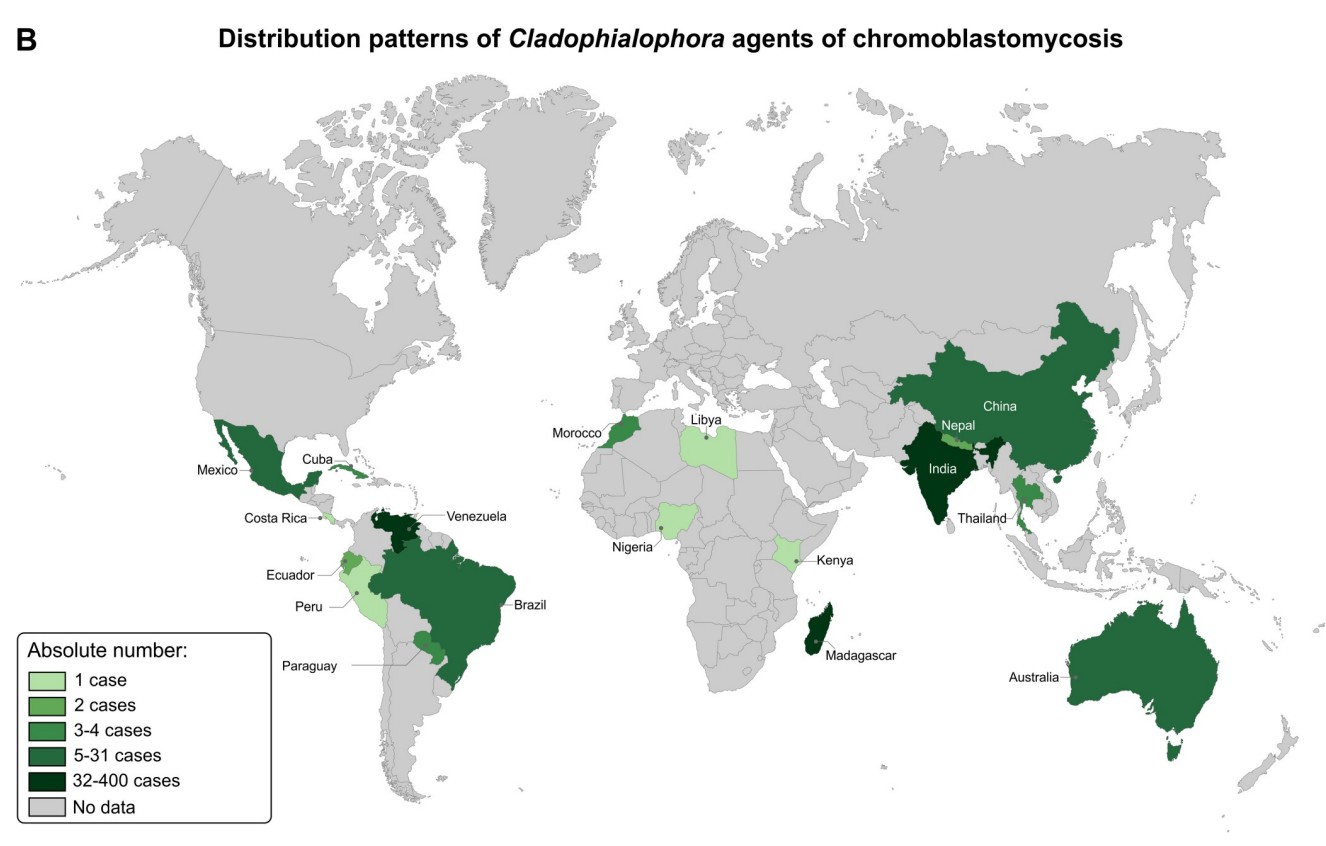

**Fig 2. The global distribution of chromoblastomycosis agents. The world map was created, edited, and colored using the vector graphics editor Corel Draw X8. Public domain link to map base layer used in creating this figure:** https://commons.wikimedia.org/wiki/File:BlankMap-World.svg.

caused by *C. carrionii* in Venezuela is underestimated in the indexed literature, specially in Falcón state, and it should be greater than 500. The highest prevalence rates of the disease per 1 million inhabitants were observed in Venezuela, French Guyana, Colombia, and Paraguay. The primary epidemiologic and clinical data of all CBM cases reported in the region were summarized in Tables 1 and S1.

## Chromoblastomycosis in Africa

A total of 65 articles were selected between 1947 and 2018 describing 1,875 patients distributed between Madagascar (1,323 cases) [94–99], South Africa (156 cases) [100–106], Republic of the Congo and Democratic Republic of the Congo (121 cases) [107–115], Gabon (64 cases) [116], Zimbabwe (35 cases) [117], Uganda (34 cases) [118], Kenya (33 cases) [119], Cameroon (23 cases) [120–123], Morocco (18 cases) [124–132], Tanzania (17 cases) [133,134], Ethiopia (14 cases) [135,136], Angola (7 cases) [115,137,138], Nigeria (5 cases) [138–141], Tunisia (5 cases) [142–146], Reunion Island (5 cases) [147–149], Libya (4 cases) [150–153], Comoro Island (4 cases) [154,155], Sierra Leone (3 cases) [156], Senegal (2 cases) [157,158], Botswana (1 case) [159] and Djibouti (1 case) [160]. There are descriptions of CBM in Chad and the Ivory Coast. Madagascar and other islands located in the Indian Ocean (Comoro and Reunion Islands) had the highest prevalence of the disease. Madagascar had 1,323 cases described in some case series published before the 1990s. New cases of CBM continue to be reported on the island, but a recent population survey has not been conducted. Data simulated by mathematical models suggest that Madagascar should have close to 2,745 cases spread throughout the country, many of them without diagnosis [161–164]. CBM is widely distributed in the central region of the continent (Democratic Republic of Congo and Republic of Congo—Congo Brazzaville), part of West Africa (Gabon), South Africa, and part of the east coast (Kenya and Tanzania) [105–116,119,133]. The disease is rarely described in desert areas.

The authors described 1,338 male and 263 female patients. Kenya, Ethiopia, Gabon, and Cameroon showed the highest male: female rates [116,119,122,135]. They mentioned the previous exposition to rural activities and history of trauma for 74.6% (129 out 173 cases) and 30.8% (20 out 65 cases), respectively. Histological findings were described for 854 (45.5%) patients. Cultures obtained from the patient's lesion yielded 1,406 fungal agents, being *Fonsecaea* spp. the most frequently found etiological agent (1,017; 72.3%), followed by *Cladophialophora* spp. (376; 26.8%). *Fonsecaea* spp. is widely distributed throughout the African continent, while *Cladophialophora* spp. is found especially in the semi-arid zones of Madagascar (366 cases) and in other countries, such as Morocco, South Africa, Lybia, and Nigeria [99,100,102,110,112,113,116,122–124,127,130,131,155]. *Phialophora* spp. was also found in some African countries at a high frequency, such as Libya, Madagascar, Kenya, South Africa, Morocco, and Djibouti [95,104,119,130,132,150,153,160].

The highest prevalence rates of CBM per 1 million inhabitants were observed in Mayotte Island, Madagascar, Gabon, and Reunion Island. The primary epidemiologic and clinical data of all CBM cases reported in the region were summarized in Tables 1 and S1.

## Chromoblastomycosis in Asia

A total of 38 articles published between 1930 and 2019 were selected. They reported 1,390 patients distributed among China (589 cases) [165], Japan (450 cases) [166–169], India (169

## A    Distribution by affected sites in the body

## B    Severity of the disease

**Face and neck**

**4.3%**

238 out of
5,555 cases

**Trunk**

**3.3%**

180 out of
5,476 cases

**Buttocks**

**1.6%**

89 out of
5,639 cases

**Upper limbs**

**19.9%**

1,120 out of
5,634 cases

**Lower limbs**

**56.7%**

3,197 out of
5,639 cases

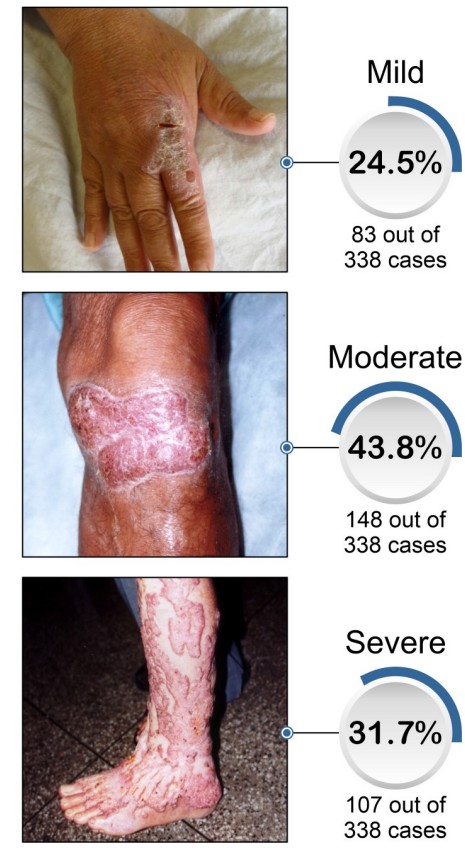

**Mild**

**24.5%**

83 out of
338 cases

**Moderate**

**43.8%**

148 out of
338 cases

**Severe**

**31.7%**

107 out of
338 cases

**Fig 3. The chromoblastomycosis lesion site.** A- The percentage of cases reported from a certain body site is shown. For lower limbs, lesions were described in 3,197 (56.7%) out of 5,639 patients; for upper limbs in 1,120 (19.9%) out of 5,634 patients; for the face and neck in 238 (4.3%) out of 5,555 patients; for the trunk in 180 (3.3%) out of 5,476 patients described; and finally for the buttocks in 89 (1.6%) out of 5,639 patients described. Unusual sites such as ear, breast, inguinal region were reported in 61 cases. B—Percentage of lesion severity.

cases) [170], Sri Lanka (71 cases) [171], Taiwan (33 cases) [172–175], Malaysia (20 cases) [176,177], Nepal (15 cases) [178,179], Thailand (14 cases) [180–183], Indonesia (13 cases) [184–187], South Korea (9 cases) [188–196], Pakistan (2 cases) [197,198], and Philippines, Bangladesh, Laos, Vietnam and Iraq each one with 1 case [199–202]. In India, the main provinces reporting cases of CBM were Kerala, Karnataka, Assam, Himachal Pradesh, and Maharashtra [170]. In Mainland China, most of the cases came from the southern provinces of Guangdong, Shandong, and Hebei [165].

The authors described 463 male and 183 female patients. Japan, Thailand, and Nepal the countries with the lowest male: female rates among patients with CBM [166–168,178,179,181,182]. They mentioned rural activities and a history of trauma for 20.3% (162 out 799 cases) and 14.3% (116 out 811 cases), respectively. Histological findings were described for 369 (26.5%) patients. Cultures from the patient's lesion allowed the identification of 750 fungal pathogens, with *Fonsecaea* spp. the most frequent etiologic agent (569; 75.8%), followed by *Cladophialophora* spp. (68; 9%) and *Phialophora* spp. (24; 3.2%) [165,167–171,174]. *Cladophialophora* spp. was found only in China, India, Thailand, and Nepal [165,170,179,182]. Less common agents, such as *Exophiala* spp., *Bipolaris* spp., *Rhinocladiella* spp., and *Rhytidhysteron* spp. were reported, especially in Japan, India, Thailand, and South Korea [167–170,183,196].

Some authors describe the latter cases are doubtful because these genera do not belong to known agents of CBM and may concern misidentifications [1,3]. Some strains of *Fonsecaea* spp. (27 strains) were subjected to molecular identification, showing *F. monophora* in 21, *F. pedrosoi* in 5, and *F. nubica* in 1 case [165,175,201]. The highest prevalence rates of CBM per 1 million inhabitants were observed in Sri Lanka, Laos, Taiwan, Japan, Malaysia, and Nepal. The primary epidemiologic and clinical data of all CBM cases reported in the region were summarized in Tables 1 and S1.

## Chromoblastomycosis in Oceania

A total of 9 articles published between 1947 and 2013 were selected, describing 168 patients from Australia (158 cases) [203–208], New Caledonia (5 cases) [209], New Zealand (4 cases) [210], and Solomon Islands (1 case) [211]. There are some reports of patients who probably acquired the disease in Samoa and the Cook Islands [209–211]. In Australia, CBM occurred predominantly in rural areas of Queensland, northern New South Wales, and the Northern Territory, including the northern part of Western Australia [205,206,209].

The authors described 133 male and 18 female patients, exhibiting one of the highest male: female rates in all studies analyzed. They mentioned the previous rural activities or a history of trauma in 40.8% (20 out 49 cases) and 42.9% (18 out 42 cases), respectively. Histological findings were described for 23 (13.7%) patients. Cultures from the lesions yielded 43 fungal agents represented by *Fonsecaea* spp. (29; 67.5%) and *Cladophialophora* spp. (14; 32.5%). *Cladophialophora* spp. was found only in Australia. *Fonsecaea* spp. was widely distributed throughout Oceania, especially in the southeast coastal area of Queensland, Australia [203,205,208–211]. The highest prevalences of CBM per 1 million inhabitants were observed in New Caledonia and the Solomon Islands. The primary epidemiologic and clinical data of all CBM cases reported in the region were summarized in Tables 1 and S1.

## Chromoblastomycosis in Europe

Excluding two publications from Russia and Finland that were written in their native languages, which precludes our analysis of data, we were able to evaluate only 35 cases of CBM published on the European continent.

The 35 European cases (24 autochthonous) were documented in the following countries: Finland (9 cases), Poland (5 cases), United Kingdom (5 cases), Czech and Slovakia (3 cases), Germany (3 cases), France (3 cases), Ukraine (2 cases), Russia (1 case), Belgium (1 case), Spain (1 case), Portugal (1 case) and Netherlands (1 case). All cases of CBM diagnosed in the UK and Netherlands, in addition to 2 cases from Germany and 1 case from France, were imported [212–218].

The authors described 27 male and 8 female patients. They mentioned the previous exposition to rural activities and history of trauma for 66.6% (2 out 3 cases) and 56.3% (18 out 32 cases), respectively. Histological findings were described for all 35 (100%) patients. Cultures from the patient's lesion yielded 30 fungal agents represented by *Fonsecaea* spp. (24; 80.1%), followed by *Exophiala* spp. (2; 6.7%), *Phialophora* spp. (1; 3.3%), *Rhinocladiella* spp. (1; 3.3%), and *Cladophialophora* spp. (1; 3.3%) [212,214–218]. The primary epidemiologic and clinical data of all CBM cases reported in the region were summarized in Tables 1 and S1.

## Chromoblastomycosis in the US and Canada

A total of 13 articles between 1915 and 2018 were selected, with 25 patients distributed between the United States (24 cases) and Canada (1 case) [219–231]. In the USA, most CBM cases were published before the 50s, and the disease is supposed to be rare nowadays. In the

USA, most CBM reports came from Massachusetts (Boston), Texas, Missouri, Michigan, Georgia, Louisiana (New Orleans), and Pennsylvania (Philadelphia) [219–230]. The single case published in Canada was probably imported once the patient had a previous history of trauma in Sri Lanka [231].

The authors described 22 male and 3 female patients. They mentioned the previous exposition to rural activities and history of trauma for 56.5% (13 out 23 cases) and 100% (3 out 3 cases), respectively. Histological findings were described for 20 (57.1%) patients. *Cultures* from the patient's lesion yielded 21 etiological agents, including *Fonsecaea* spp. (15; 71%), and *Phialophora* spp. (6; 29%). *Cladophialophora* spp. was not reported in any case from the USA [219–231]. The primary epidemiologic and clinical data of all CBM cases reported in the region were summarized in Tables 1 and S1.

## Consolidated worldwide CBM data

A total of 7,740 cases of CBM was described in five continents. The authors described 4,022 (81.7%) male and 896 (18.3%) female patients. The median age was 52.5 years (range between 2–93 years), and the average time between the onset of the first lesion and CBM diagnosis was 9.2 years (range between 1 month to 50 years). The authors mentioned exposure to rural activities and history of trauma for 56.8% (1,895 out 3,334 cases) and 33.1% (568 out 1,717 cases), respectively. Histological findings were described for 2,125 (27.8%) patients.

The presence of immunosuppressive diseases at the time of diagnosis of CBM was reported in only 16 (0.2%) cases, with solid organ transplantation the most common condition (kidney, heart transplantation), followed by HIV infection, rheumatoid arthritis, systemic lupus erythematosus, bladder neoplasia, celiac disease, pernicious anemia, and non-Hodgkin lymphoma [66,68,92,148,170,229]. Concomitant infection diseases were reported in 19 cases of CBM, including mycetoma (5 cases), leprosy (5 cases), cutaneous filariasis (3 cases), paracoccidioidomycosis (2 cases), cutaneous histoplasmosis (1 case), syphilis (1 case), actinomycosis (1 case) and visceral leishmaniasis (1 case) [19,50,54,56,60,65,68,108,121,157,158,170].

CBM lesions were present in only one body segment in 1,313 out 1,472 cases (89.2%) and in more than one body segment in 159 out 1,472 cases (10.8%). Itching and pain were reported by 21.5% (281 out 1,309 cases) and 11% (145 out 1,313 cases), respectively.

The main sites involved were the lower limbs (3,197 out 5,639 cases; 56.7%), followed by the upper limbs (1,120 out 5634 cases; 19.9%), head and neck (158 out 5,555 cases; 2.9%), trunk (180 out 7,568 cases; 2.4%), buttocks (89 cases) and unusual sites such as ear, breast, inguinal region (61 cases).

The main patterns of dermatological lesions were verrucous (710 cases), followed by tumorous (663 cases), plaque (565 cases), nodular (285 cases), ulcers (105 cases), and scarring (75 cases) lesions. Approximately 63.2% (470 out 743) of the patients had only 1 pattern of dermatologic lesions, while polymorphisms of lesions were observed in 36.8% (273 out 743) of patients. Regarding the severity of the disease by Carrión's criteria, 83 out 338 patients (24.5%) had mild, 148 out 338 patients (43.8%) moderate, and 107 out 338 patients (31.7%) severe forms. Malignant transformation diagnosed by histopathology of CBM lesions was described in 22 cases. Except for one case of melanoma, all others were described as squamous and basal cell carcinomas. A total of 3,817 fungal isolates were cultured, being 3,089 (80.9%) *Fonsecaea* spp., 552 (14.5%) *Cladophialophora* spp., and 56 *Phialophora* spp. (1.5%). The primary epidemiologic and clinical data of all CBM cases reported in the world were summarized in Figs 1–3 and Tables 1 and S1.

Data about the treatment of CBM were scarcely documented, with incomplete information in most papers. In this regard, the management of lesions solely by surgical excision or

physical methods (cryotherapy, thermotherapy, and photodynamic therapy) was documented in 133 cases. Surgical debridement as adjuvant therapy was described in additional 191 cases [35,37,39,47,50,100,110,113,165,170,182,204,212]. Itraconazole was the antifungal therapy mostly used in all continents, being reported in 318 patients, followed by treatment with terbinafine that was used by 87 patients, especially in India, China, and Madagascar [18,65,67,68,70,77,99,165,170]. The use of fluconazole, ketoconazole, flucytosine, and systemic or intralesional amphotericin were only sporadically reported. Curiously, the administration of iodides as pharmacological therapy was described in 103 cases, and it was a common practice in India, Cuba, and Australia [19,170,204,212,228]. Unfortunately, clinical and laboratory follow up data were not available to check for the clinical response (complete or partial cure) in the mentioned articles. A total and partial cure were described in 302 and 188 cases, respectively. Amputation of the affected limb was observed in 14 cases [56,67,68,76,99,103,116,119,167,181].

## Discussion

The true burden of CBM is not known. A lack of national surveillance systems checking for CBM in sentinel centers does not exist [1,2,232,233]. This paper represents the most comprehensive review of CBM cases published between 1914 and 2020, providing data to partially characterize the relative burden of this neglected implantation mycosis in different countries and the main clinical and mycological characteristics of the affected patients.

Our review showed that CBM has been widely described on all continents over the last eight decades and thrives in areas where access to adequate sanitation, clean water, and healthcare is limited. Regardless of the country considered, CBM is diagnosed in people who live in remote and rural areas and affects some of the world's poorest and most marginalized communities, predominantly in Africa, Asia, and America [1,2,232]. Rural areas in developing countries highly endemic for CBM generally present high informal employment arrangements, low human development index, and lack of appropriate social protection systems for agriculture workers. In most countries, surveillance practices for personal protective equipment (PPE) in agriculture are unknown, and their use in rural areas is woefully inadequate and requires more attention. The lack of protective shoes, gloves, or garments associated with poor hygienic habits and insufficient nutrition may favor development of CBM after infection by implantation [1–7,14,234–236]. It is not known if there are other factors affecting the development of CBM in particular individuals or disease expression and severity. For example, an inability of toll like receptor 7 (TLR7) to recognize and respond appropriately to the causative fungi could underly the progressive nature of CBM in some patients [237].

Although the route of acquisition of CBM agents is by traumatic inoculation, most of the series did not track the source of infection once clinical manifestations as several years usually elapse after trauma when the patients and the lesion(s) grows very slowly. We were not capable to analyze data related to trauma or characterization of rural jobs as this information were not available in most papers [1,3,6,18,19,27,35,56,70,116,165,238,239].

Rarely CBM is described simultaneously with other NTDs (mycetoma, leprosy, filariasis and leishmaniasis). This occurrence reflects areas of co-endemicity, with common environmental exposure in populations under conditions of poverty [1,3,54,56,57,60,108,121].

Although the diagnosis of CBM does not rely on expensive and sophisticated laboratory tools, the disease remains neglected by all health systems, making the time of diagnosis too long (mean of 9 years). This aspect certainly impacts in morbidity, including disease progression, risk of superinfection and malignant transformation [8,56,68,99,167,175,240,241,242].

Considering the insidious progression of the fungal disease, patients continue their labor and social activities for many years before having the diagnosis made [1,3,5,48,176].

Notably, the disease is mostly observed in males probably due to different environmental exposition and possible protection of women by endogenous steroids [1,5]. This hypothesis needs further investigation and validation. However, in some countries the prevalence in women is high probably due to their involvement in agricultural activities [16,43,44,116,122,166–168,199]. Lower limbs are the most common site affected, although some countries frequently reported lesions in upper limbs due to local practice of carrying wood or other agricultural materials in arms or shoulders [16,19,20,165,180,182,204,205,228,229].

In the present review, we adopted the CBM Carrión classification for skin lesions considered the most consistent and comprehensive description of dermatological lesions with updated nomenclature [1,3,65,68]. As expected, warty lesions (29%), tumors (27%), infiltrative plaques (23%), and nodules (12%) were the most common pattern of CBM, but polymorphic lesions may also be found, especially in patients with a long history and chronic evolution of the process [52,56,65,67,68,99,165,169,175]. The main symptoms were itching (21%) and pain (11%), with local edema rarely reported by most authors [6,7,9,21,28,30]. Of note, the pattern of skin lesions is not linked to the etiological agent of CBM.

Some series have shown that secondary bacterial infections and lymphedema are concerns. Uncommonly, malignant transformation may occur, especially in patients with a long history of CBM diagnosis [1,3,54,67,68,76,99,103,116,119,167,240–242]. CBM progresses slowly, produces fibrotic changes and lymphatic stasis. Secondary recurrent bacterial infection exacerbates the involvement of lymphatic vessels, resembling elephantiasis. Severe forms of CBM disable and disfigure patients much more frequently than they kill, and are multifactorial [243,244]. Affected people live decades with disability, stigma and social withdrawal. Disability Adjusted Life-Years (DALY) lost due to CBM has not been comprehensively evaluated in endemic areas [1,3–5,7,14,15].

Laboratory diagnosis of CBM requires only the visualization of single or clustered muriform cells by direct mycological examination or histopathology. Most diagnoses in North American and European countries are provided by histological examination. In contrast, in Asian, African, and Latin American countries where CBM is endemic, less than 30% of all cases required biopsy for the diagnosis [8,16,25,49,51,56,68,99,169]. Direct mycological examination with potassium hydroxide solution of skin scrapings containing crusts or cellular debris is a fast and straightforward tool frequently useful in low-income countries to diagnose patients with CBM [56,68,99,170].

Although the different agents of CBM have certain ecological features, there is no apparent impact of the diversity of species on clinical manifestations or therapeutic management. Molecular characterization of different species has mostly been used to characterize this ecology and epidemiological aspects of the various etiological agents of CBM [1,11–13,238,239].

*Fonsecaea* spp. is the main genus causing CBM worldwide [16,18–25,50–56,68,99,116,165,169,170,189–191]. Molecular studies showed that *Fonsecaea pedrosoi* is the main species within this genera, and it is found practically in all countries where CBM has been reported. This species causes almost exclusively subcutaneous disease, with rare visceral involvement [11,12,16,68]. Disseminated forms of the disease have also been reported but without unambiguous muriform cells in tissue (44), and thus, they may be considered phaeohyphomycosis. *Fonsecaea monophora* is widely distributed, with high prevalence, especially in Asia and subtropical or temperate countries. *F. monophora*, together with *F. pugnacius*, can cause disseminated CBM or phaeohyphomycosis with visceral impairment [165,245,246].

Finally, *Fonsecaea nubica* is also widely distributed in Asia, but current studies showed that Madagascar might be the country with the highest number of CBM caused by this species [239].

The second main etiological agent of CBM is *C. carrionii*, with Venezuela, Madagascar, Australia, India, and China the countries most affected [44,45,48,49,99,165,170,203–208]. This agent is typically found in arid and semi-arid climates, with average yearly temperatures of 24˚C, scarce rainfall (up to 600 annual mL) and is located at moderate altitude (up to 500 m) [43,48,99,208]. Finally, *Phialophora* spp. (*P. verrucosa*) is an uncommon agent, responsible for almost 30% of cases in the USA [166–170,219–221,228,230]. *Rhinocladiella* spp. is the etiologic agent of less than 1% of CBM cases [16,68,212].

Treatment of CBM is difficult, and several different therapeutic regimens have been tried, including physical methods. Most of small initial lesions in mild disease can be excised surgically, but clinical data and follow-up of these patients are incompletely characterized. CBM lesions are refractory, and healing is almost impossible to achieve, especially in its moderate to severe clinical presentations [19,21,22,30,52,95,99]. Although there are no randomized clinical trials to define the best choice for its treatment, itraconazole is the main antifungal drug used, specially 400 mg per day in moderate and severe cases, based on observational studies [65,66,68]. Terbinafine is the second most frequently used antifungal drug, specially in some countries as Madagascar and China, based on open and non-comparative clinical trials [163,240]. Voriconazole, posaconazole, and isavuconazole are used only in refractory disease [1,247,248]. Interestingly, some therapies have been abandoned, such as cholecalciferol, thiabendazole, intravenous amphotericin, ketoconazole, and topical 5-fluorouracil. Due to the low cost, potassium iodide has been used in some countries, especially in Cuba and India [19,23,170]. Adjuvant therapy for improve the cellular immune response with topical imiquimod or intramuscular glucan was used mostly in more severe and refractory cases [249–251].

## Conclusions

Despite all limitations, our study provides a comprehensive review of clinical and therapeutic aspects of CBM and an estimate of the prevalence of the disease in each country. Our maps have shown CBM to be widespread in five different continents, specially in Latin America, Africa and Asia. Countries such as Madagascar, Gabon, Indian Ocean Islands (Comoro and Reunion), Costa Rica, Dominican Republic, Venezuela, French Guiane, and Island of Oceania (New Caledonia) are the countries with the highest incidence densities in the world. CBM in world is probably more common than expected. The disease especially affects men (81.7%), with an average delay of 9.2 years between onset and diagnosis. The mean age was 57.1 years (range 2–93 years), being the lower and upper limbs the most compromised sites. Verrucous, tumoral and plaque represent the main dermatological patterns. *Fonsecaea* spp. is the main etiological agent, being widely distributed on all continentes and responsible for more than 80% of cases. This review allows the understanding of a gap in epidemiological, diagnostic and therapeutic data. There is an urgent need to create and implement social protection policies for vulnerable populations and national programs for the diagnosis and treatment of the disease.

## Supporting information

**S1 Table. Rate of occurrence, clinical and demographic data of chromoblastomycosis cases stratified by countries.**
(DOCX)

## Acknowledgments

We thank all the Brazilian Network of Melanized Fungal Infections and the Working Group of Chromoblastomycosis of ISHAM for the encouragement in the development of this work.

## Author Contributions

**Conceptualization:** Daniel Wagner C. L. Santos, Conceição de Maria Pedrozo e Silva de Azevedo, Vania Aparecida Vicente, Flávio Queiroz-Telles, G. Sybren de Hoog, David W. Denning, Arnaldo Lopes Colombo.

**Data curation:** Daniel Wagner C. L. Santos, Arnaldo Lopes Colombo.

**Formal analysis:** Daniel Wagner C. L. Santos, Vania Aparecida Vicente, Flávio Queiroz-Telles, Anderson Messias Rodrigues, G. Sybren de Hoog, David W. Denning, Arnaldo Lopes Colombo.

**Investigation:** Daniel Wagner C. L. Santos, Arnaldo Lopes Colombo.

**Methodology:** Daniel Wagner C. L. Santos, Conceição de Maria Pedrozo e Silva de Azevedo, Flávio Queiroz-Telles, Arnaldo Lopes Colombo.

**Project administration:** Daniel Wagner C. L. Santos, Arnaldo Lopes Colombo.

**Resources:** Daniel Wagner C. L. Santos.

**Software:** Daniel Wagner C. L. Santos, Anderson Messias Rodrigues.

**Supervision:** Daniel Wagner C. L. Santos, Conceição de Maria Pedrozo e Silva de Azevedo, Vania Aparecida Vicente, Flávio Queiroz-Telles, G. Sybren de Hoog, David W. Denning, Arnaldo Lopes Colombo.

**Validation:** Daniel Wagner C. L. Santos, Conceição de Maria Pedrozo e Silva de Azevedo, Vania Aparecida Vicente, Flávio Queiroz-Telles, Anderson Messias Rodrigues, G. Sybren de Hoog, David W. Denning, Arnaldo Lopes Colombo.

**Visualization:** Daniel Wagner C. L. Santos, Flávio Queiroz-Telles, Anderson Messias Rodrigues, David W. Denning, Arnaldo Lopes Colombo.

**Writing – original draft:** Daniel Wagner C. L. Santos, Arnaldo Lopes Colombo.

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
