## [Decision Letter · Decision Letter 0]

17 May 2021

Dear colleague

Thank you very much for submitting your manuscript "The global burden of chromoblastomycosis." for consideration at PLOS Neglected Tropical Diseases. As with all papers reviewed by the journal, your manuscript was reviewed by members of the editorial board and by several independent reviewers. The reviewers appreciated the attention to an important topic. Based on the reviews, we are likely to accept this manuscript for publication, providing that you modify the manuscript according to the review recommendations. 

Sincerely,

Roderick Hay

Guest Editor

Olivier Chosidow

Deputy Editor

Reviewer's Responses to Questions

**Key Review Criteria Required for Acceptance?**

**Methods**

-Are the objectives of the study clearly articulated with a clear testable hypothesis stated?

-Is the study design appropriate to address the stated objectives?

-Is the population clearly described and appropriate for the hypothesis being tested?

-Is the sample size sufficient to ensure adequate power to address the hypothesis being tested?

-Were correct statistical analysis used to support conclusions?

-Are there concerns about ethical or regulatory requirements being met?

Reviewer #1: The research carried out has a clear method and the results obtained are consistent, which does allow for an impact of the CBM

Reviewer #2: The manuscript was presented as a review of chromoblastomycosis, a topic of great concern and much merit. In general, the body of the manuscript is literally consistent with the title. Breaking up this manuscript into several continents has also provided a reader-friendly text.

Reviewer #3: This review article aims to understand the global burden of Chromoblastomycosis(CBM)-an important neglected tropical disease(NTD). The authors retrospectively conducted a comprehensive systematic review of medical literature published during the past 106 years in four different languanges. A total of 7,740 cases of CBM were described in five continents. 

The prevalence rates, geographic distribution, and clinical aspects of CBM were analyzed.

**Results**

-Does the analysis presented match the analysis plan?

-Are the results clearly and completely presented?

-Are the figures (Tables, Images) of sufficient quality for clarity?

Reviewer #1: The results obtained, analysis and meta-analysis are adequate and the support of the tables and figures are adequate. Here I suggest that you add a human figure indicating (average) of the clinical locations that will make the information clearer

Reviewer #2: Furthermore, epidemiological, clinical and mycological data, and therapeutic options were discussed accordingly. Epidemiological and mycological (distribution patterns of the genera Fonsecaea and Cladophialophora) characteristics were also implemented using clean and clear Figures (resolutions should be checked) as well a Table.

Reviewer #3: This study provides a comprehensive review of clinical and therapeutic aspects of CBM and an estimate of the prevalence of the disease in each different area of the world. The analysis results were clearly presented. 

However, the Table 1 was not meeting the standard requirement of a table (better us four lines formed table).

The map of each result was not very clear, especially in the agents distribution, the absulute number was not correct. For example, in China, C. carrionii took 60% of the pathogenic fungi, at least over 300 cases caused by this fungus.

**Conclusions**

-Are the conclusions supported by the data presented?

-Are the limitations of analysis clearly described?

-Do the authors discuss how these data can be helpful to advance our understanding of the topic under study?

-Is public health relevance addressed?

Reviewer #1: They are adequate according to the results obtained

Reviewer #2: The authors have addressed all relevant queries regarding CMB.

Reviewer #3: Yes, the conclusions were supported by the data analysis and the authors discussed the distribution of CBM, the pathogenic fungi, clinical types and the treatment and prognosis. The limitations of the literatures they reviewed were mentioned. They discussed how these data could be helpful to advance our understanding of the disease burden of CBM and the public health relevance was addressed, too.

**Editorial and Data Presentation Modifications?**

Reviewer #1: My suggestions are attached to the editor and the authors

Reviewer #2: After reading carefully the manuscript I listed some topics for the authors consideration:

Lines 43, 404, and 498. Malignant transformation, any specific data/type?

Line 131, dosis 

Line 191, absolut

Line 360, Exophila

Line 469, discrebed 

Line 480, deference

Line 483, envolvement 

Line 469, discrebed, typos..

Line 498, long history? Any prediction, months/years?

Reviewer #3: Minor revisions are needed:

Data presentation: 

1. The distribution maps, number of the pathogen.

2. Table 1, the forms should be modified to four lines.

**Summary and General Comments**

Reviewer #1: It is an extraordinary piece of research that could be accepted as is. It gives a clear impact of the CMB in the world and provides much more precise data.

My only suggestions:

The second group should be called Central America, the Caribbean and Mexico.

I suggest that similar to the work of Van de Sande W (Ref 2), which was the model, they integrate a figure with the main locations, that leaves more accessible information on the clinical topography

Reviewer #2: The authors have produced a nice review of chromoblastomycosis with evidence across the globe. The list of demographic, epidemiological, and clinical aspects leads it to a leading article in the field.

Reviewer #3: This paper provides the most comprehensive review of the literatures published during 1914~2020, which summarized the clinical and therapeutic aspects of CBM. Through the estimate of the burden of the disease, showing CBM to be widespread, specially in Latin America, Africa and Asia. The ratio of the male to female was very high and the disease diagnosis was terribally delayed. The most infected sites and clinical patterns were analyzed. Regards to the pathogenic fungi, Fonsecaea spp. was the agent of more than 80% of cases. This review allows the better understanding of the epidemiological picture, and the diagnostic and therapeutic status of CBM. Though this review, we could understand the necessay of the improvement of the diagnosis and treatment of CBM and the protection of the people to aviod the infction. 

The limitation of the review is due to the literatures were neither high quality epidemiological study nor comprehensive surveillancethe. For example, the diagnosis criteria may not meet togethor, if the phaeohyphomycosis was included or not?

The risk factors were not included, such as the immunocomprised patients, if the genetic susceptible patients such as CARD9 deficiency were inculded?

PLOS authors have the option to publish the peer review history of their article (what does this mean?). If published, this will include your full peer review and any attached files.

Reviewer #1: Yes: Alexandro Bonifaz

Reviewer #2: Yes: Macit Ilkit

Reviewer #3: No

Figure Files:

Data Requirements:

Reproducibility:

References

---

## [Decision Letter · Decision Letter 1]

30 Jun 2021

Dear Dr Santos

We are pleased to inform you that your manuscript 'The global burden of chromoblastomycosis.' has been provisionally accepted for publication in PLOS Neglected Tropical Diseases.

Best regards,

Roderick Hay

Guest Editor

Olivier Chosidow

Deputy Editor

Reviewer's Responses to Questions

**Key Review Criteria Required for Acceptance?**

**Methods**

-Are the objectives of the study clearly articulated with a clear testable hypothesis stated?

-Is the study design appropriate to address the stated objectives?

-Is the population clearly described and appropriate for the hypothesis being tested?

-Is the sample size sufficient to ensure adequate power to address the hypothesis being tested?

-Were correct statistical analysis used to support conclusions?

-Are there concerns about ethical or regulatory requirements being met?

Reviewer #1: If they are suitable and correct for the type of work

Reviewer #2: This section is clearly presented.

**Results**

-Does the analysis presented match the analysis plan?

-Are the results clearly and completely presented?

-Are the figures (Tables, Images) of sufficient quality for clarity?

Reviewer #1: The results obtained are adequate, the figures, maps and tables are adequate in shape and number.

Reviewer #2: The results are presented in a reader-friendly format.

**Conclusions**

-Are the conclusions supported by the data presented?

-Are the limitations of analysis clearly described?

-Do the authors discuss how these data can be helpful to advance our understanding of the topic under study?

-Is public health relevance addressed?

Reviewer #1: Yes, it is appropriate based on the results

Reviewer #2: This section is organized adequately.

**Editorial and Data Presentation Modifications?**

Reviewer #1: None

Reviewer #2: This reviewer believes that this version is ready for publication.

**Summary and General Comments**

Reviewer #1: Adequate

Reviewer #2: Overall, the manuscript is sound and will be a leading paper of this field.

PLOS authors have the option to publish the peer review history of their article (what does this mean?). If published, this will include your full peer review and any attached files.

Reviewer #1: **Yes: **Alexandro Bonifaz

Reviewer #2: **Yes: **Macit Ilkit

---

## [Editor Report · Acceptance letter]

23 Jul 2021

Dear Dr Santos,

We are delighted to inform you that your manuscript, "The global burden of chromoblastomycosis.," has been formally accepted for publication in PLOS Neglected Tropical Diseases.

Best regards,

Shaden Kamhawi

co-Editor-in-Chief

Paul Brindley

co-Editor-in-Chief
